# A Mallows-like criterion for anomaly detection with random forest implementation

**GaoXiang Zhao[1], Lu Wang[2], Xiaoqiang Wang** [2]*

1 Department of Mathematics, Harbin Institute of Technology (Weihai), Weihai, Shandong, China,
2 Department of Mathematics and Statistics, Shandong University, Weihai, Shandong, China

* xiaoqiang.wang@sdu.edu.cn

## Abstract

Anomaly detection plays a crucial role in fields such as information security and industrial production. It relies on the identification of rare instances that deviate significantly from expected patterns. Reliance on a single model can introduce uncertainty, as it may not adequately capture the complexity and variability inherent in real-world datasets. Under the framework of model averaging, this paper proposes a criterion for the selection of weights in the aggregation of multiple models, employing a focal loss function with Mallows' form to assign weights to the base models. This strategy is integrated into a random forest algorithm by replacing the conventional voting method. Empirical evaluations conducted on multiple benchmark datasets demonstrate that the proposed method outperforms classical anomaly detection algorithms while surpassing conventional model averaging techniques based on minimizing standard loss functions. These results highlight a notable enhancement in both accuracy and robustness, indicating that model averaging methods can effectively mitigate the challenges posed by data imbalance.

## 1 Introduction

Anomaly Detection (AD) [1–3] aims to identify anomalies from observed data. Its applications include financial analysis [4], cybersecurity [5], and paramedical care [6]. Traditional anomaly detection methods, such as Isolation Forest (IF), the Local Outlier Factor (LOF), and Gaussian Mixture Models (GMMs), often assume that anomalies are outliers or low-probability points, and they distinguish anomalies by attributes based on statistical properties and distance density [7].

These assumptions can be subject to two types of limitations. First, the inherent uncertainty associated with any single model can cause fitting issues. Second, detection is confined to a singular type of anomaly, which can be inefficient, especially when the true anomaly type differs significantly from that of the method.

Ensemble methods can account for different types of data and models, but conventional vote or mean approaches may prove inefficient for extremely imbalanced data. Model average methods [8–10] have been confirmed to improve overall prediction performance by com-

**Data availability statement:** All datasets are available at https://zenodo.org/records/15226740 (DOI: https://doi.org/10.5281/zenodo.15226740).

**Funding:** The author(s) received no specific funding for this work.

**Competing interests:** The authors have declared that no competing interests exist.

bining the prediction results of multiple base models. Such methods are based on a minimum loss function [11,12], and some are Bayesian [13]. Focal loss variational autoencoders have been combined with the XGBoost method, and this combination has achieved promising results when applied to imbalanced network traffic datasets. Other work has focused on feature selection methods for intrusion detection, as well as deep learning approaches and lightweight designs tailored for intrusion detection systems. Some researchers have intensively investigated model averaging methods for high-dimensional regression problems by removing weight constraints or handle the problem in the presence of responses missing at random, while others have investigated the improvement of Bayesian model averaging methods, such as combining a Bayesian model with the selection of regressors, to enhance interpretability and efficiency of Bayesian model averaging estimation. Some researchers have also explored the application of model averaging in the field of deep learning and proposed an efficient protocol for decentralized training of deep neural networks from distributed data sources, which has achieved good results in several deep learning tasks. In the model averaging method based on minimizing the loss function, the loss function is usually selected by choosing the logarithmic loss function, the squared loss function, and the cross-entropy loss function. [14–17]. The focal loss function [18] addresses class imbalance, particularly in object detection and image segmentation. It can assign different weights to samples or classes, but its application to model averaging, specifically in assigning weights to base models, has not been explored. Model averaging using Mallows' criterion is known for its asymptotic optimality in linear regression problems and has been extended to some machine learning models, such as the random forest. However, model averaging has primarily focused on regression problems, with the goal of predicting continuous outcomes. While this has improved prediction accuracy and stability in regression tasks, it has seen little application to classification problems. This is problematic for datasets that show significant class imbalance. We aim to extend model averaging methods to better meet these challenges. Specifically, we adapt the Mallows criterion by substituting the conventional cross-entropy loss function with a focal loss function. This enables our ensemble of submodels to learn more effectively from minority classes without compromising performance on majority classes.

We propose optimizing the weights in model averaging by integrating a focal loss function into the Mallows criterion. Specifically, within a random forest framework, we introduce a complexity penalty term to the focal loss function, akin to Mallows averaging [19], and determine the weights for sub-decision trees by minimizing a Mallows-like criterion. This approach enhances performance on highly imbalanced datasets through the use of the focal loss function, which improves anomaly detection accuracy. It also controls model complexity and boosts generalization by incorporating a regularization term into the loss function and leverages model averaging to amalgamate the strengths and performance of various base models, assigning them distinct weights. Consequently, this method facilitates the development of a more precise anomaly detection model.

The proposed Mallows-like focal loss approach is compared with anomaly detection methods based on minimizing other loss functions, as well as commonly employed anomaly detection methodologies. The proposed methodology is evaluated using the AUC to assess binary classification performance, ARI to assess clustering algorithm performance, and the recall metric to assess the percentage of outliers that are detected. Evaluated on the KDDCup network intrusion dataset, the proposed approach shows a 2.90% improvement in the F1-score over the suboptimal model averaging method based on minimizing the cross-entropy loss function and a 5.75% improvement of the recall metric. Our approach also shows superior performance to several common anomaly detection methods. Public benchmark datasets

were also employed, with results indicating improved accuracy and stability in anomaly detection and extremely imbalanced data classification.

The main contributions of this paper are summarized as follows:

- We propose a Mallows-like averaging criterion to optimize the weights in the aggregation of multiple models; in particular, the focal loss function is instrumental in enhancing the performance of anomaly detection;
- Utilizing Mallows-like focal loss (MFL), we introduce a variant of the random forest algorithm, tailored for anomaly detection, within the framework of model averaging for optimal weight selection.

## 2 Proposed method

We consider anomaly detection as a binary supervised classification problem. We summarize the training sample as a set, $D = \{(Y_1, \boldsymbol{x}_1), \cdots, (Y_n, \boldsymbol{x}_n)\} \subseteq \{0, 1\} \times \mathbb{R}^p$, where $\boldsymbol{x}_i = (x_{i1}, \cdots, x_{ip})^\top$ is a vector of predictors of dimension $p$; the response variable $Y_i = 1$, $i = 1, \cdots, n$, indicates that the $i$-th sample is abnormal, and otherwise its value is zero. The relationships between all variables can be formulated as

$$Y_i = f(\boldsymbol{x}_i) + \varepsilon_i, \quad i = 1, \cdots, n, \tag{1}$$

where $f(\cdot)$ is unspecified, or even nonparametric. We suppose that all the residual errors $\varepsilon_i$ are independent and homogeneous, with $E(\varepsilon_i) = 0$, $E(\varepsilon_i^2) = f(\boldsymbol{x}_i)(1 - f(\boldsymbol{x}_i))$.

Notice that in Model (1), the $p$ predictors $\boldsymbol{x}$ can be randomly selected, thereby generating diverse models. Model averaging involves the weighted ensemble of models corresponding to each variable selection, i.e.,

$$f(\boldsymbol{x}_i) = \sum_{m=1}^{M} \omega_m f_m(\boldsymbol{x}_i), \tag{2}$$

followed by the minimization of a penalized loss function to optimize the selection of optimal weights within the unit simplex,

$$\mathcal{H} = \left\{ \omega = (\omega_1, \cdots, \omega_M) \in [0, 1]^M \mid \sum_{m=1}^{M} \omega_m = 1 \right\}. \tag{3}$$

The focal loss is initially designed to address the object detection scenario for extremely imbalanced data, adding a modulating factor to the standard cross-entropy criterion,

$$\begin{aligned} FL(Y, f(\boldsymbol{x})) = &- Y\alpha(1 - f(\boldsymbol{x}))^\gamma \log(f(\boldsymbol{x})) \\ &- (1 - Y)(1 - \alpha)f(\boldsymbol{x})^\gamma \log(1 - f(\boldsymbol{x})), \end{aligned} \tag{4}$$

where $f(\boldsymbol{x})$ is the estimated probability for the class with label $Y = 1$. Incorporating focal loss into the ensemble method, we propose a Mallows-like criterion to determine the optimal weights. This Mallows-like criterion is achieved through a random forest algorithm. Fig 1 shows the framework of the proposed method, which is a model averaging structure based on minimizing the MFL criterion.

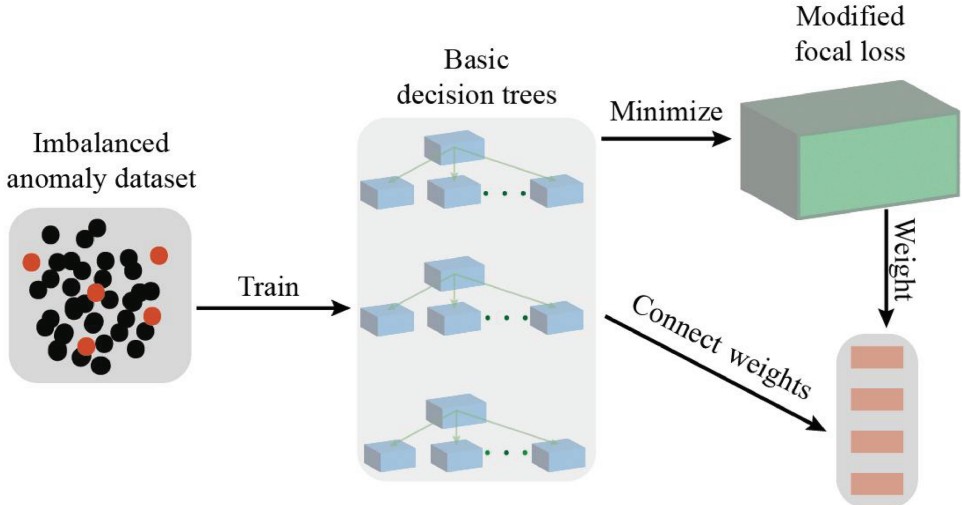

**Fig 1. Schematic diagram of proposed model averaging method.** This minimizes the MFL criterion to allocate weights to base decision trees, mitigating the effects of data imbalance while controlling model complexity.

## 2.1 Mallows-like focal loss criterion

Hansen first investigated the Mallows criterion in least squares model averaging, selecting the weight vector as

$$\omega^* = \arg\min_{\omega \in \mathcal{H}} \left\{ ||Y - P(\omega)Y||^2 + 2\sigma^2 \text{trace}\left(P(\omega)\right) \right\}, \tag{5}$$

where $P(\omega) = \sum_{m=1}^{M} \omega_m P_m$, $\sigma^2$ is an unknown parameter to estimate, and $P_m$ is the projection matrix in linear regression for the $m$-th model. The term $\text{trace}(P(\omega))$ is defined as the effective number of parameters, i.e., the weighted average of the number of predictors in each submodel. The optimized function has two terms, the first measuring the fitting error of the weighted model, and the second penalizing model complexity.

By substituting the first term in the Mallows criterion with focal loss (4), we realize a criterion for anomaly detection,

$$C(\omega) = \sum_{i=1}^{n} FL\left(Y_i, f(x_i)\right) + 2\sigma^2(\omega)\left(1 + \sum_{m=1}^{M} \omega_m k_m\right), \tag{6}$$

where $k_m$ is the number of predictors in the $m$-th base model. Depending on the implemented algorithm, $\sum_{m=1}^{M} \omega_m k_m$ can be relaxed to a function of the number of predictors in each base model.

Considering the random forest method and Least Squares Support Vector Classification (LSSVC), in the random forest, $k_m$ is the number of internal nodes within the $m$-th trained decision tree, and in LSSVC, it measures the magnitude of support vector weights. For the $m$-th classifier, $k_m$ takes the form

$$\text{trace}\left(\left(HH^\top + \lambda I\right)^{-1} HH^\top\right), \tag{7}$$

where the matrix $H$ represents the two-dimensional mapping of support vectors relative to the entire sample set via the kernel trick, and $\lambda$ indicates the strength of regularization.

Note that the unknown parameter $\sigma^2(\boldsymbol{\omega})$ in (6) represents the variance of the model in the Mallows criterion, which is replaced by $\hat{\sigma}^2(\boldsymbol{\omega}) = \sum_{i=1}^{n} FL(Y_i, f(\boldsymbol{x}_i))/n$ in practice.

## 2.2 Random forest with MFL criterion

The random forest is a popular classification method due to its flexibility and accuracy, where the voting mechanism is frequently utilized for data classification. By contrast, the first term in the MFL criterion (6) measures the fitting error of the weighted random forest in the training sample. The second term in (6) penalizes the complexity of trees in the forest, where $\sum_{m=1}^{M} \omega_m k_m$ is the weighted number of leaf nodes of all trees.

Utilizing the MFL criterion (6), we realize this algorithm as follows. We establish $M$ decision trees, $\hat{f}_m(\boldsymbol{x})$, $m = 1, \cdots, M$, and apply the Mallows-like criterion to optimize the weight, which is denoted by $\omega^*$. The weighted base models are linearly combined to form the overall model. Algorithm 1 shows the steps of the model averaging method.

**Algorithm 1. Random forest with Mallows-like focal loss criterion.**

**Require:** Dataset $D = \{(\boldsymbol{x}_i, Y_i), i = 1, \cdots, n\}$, number of subtrees $M$, hyperparameters $\alpha, \gamma$.

**Ensure:** Optimal weights $\boldsymbol{w}^*$

  **while** decision tree $m = 1, \cdots, M$ **do**
     Create a bootstrap sample $D_m$ of size $n$ from the training data $D$
     Randomly select $[\sqrt{M}]$ predictors
     Build a decision tree $\hat{f}_m$ with $k_m$ leaf nodes
  **end while**
  Optimize the weights $\boldsymbol{\omega}$

$$\omega^* = \min_{\omega \in \mathcal{H}} \left\{ \sum_{i=1}^{n} FL(Y_i, f(\boldsymbol{x}_i)) + 2\hat{\sigma}^2(\boldsymbol{\omega}) \left( 1 + \sum_{m=1}^{M} \omega_m k_m \right) \right\}$$

  with

$$f(\boldsymbol{x}_i) = \sum_{m=1}^{M} \omega_m \hat{f}_m(\boldsymbol{x}_i),$$

$$\hat{\sigma}^2(\boldsymbol{\omega}) = \sum_{i=1}^{n} FL(Y_i, f(\boldsymbol{x}_i))/n.$$

There are two hyperparameters, $\alpha, \gamma$, in focal loss function (4), and there are $M$ subtrees in the random forest algorithm. We adopt Bayesian hyperparameter estimation methods [20] to expedite training and optimize the results. As the focal loss function represents a nonlinear constrained optimization problem, we employ sequential least squares [21] to optimize the modified focal loss function, thereby controlling complexity and computational costs.

## 3 Experiments

### 3.1 Main result

Anomaly detection typically addresses the issue of severe imbalance, where there is often no clear definition regarding the proportions of positive and negative samples. To expedite

training and improve the positive-to-negative sample ratio for validating model effectiveness in anomaly detection, we employed simple random sampling for the minority class while controlling the ratio of positive to negative samples to be 0.05.

We first consider the publicly available KDDCup network intrusion dataset, which is used in the field of anomaly detection, whose 125, 870 records each have 41 features. We proportionally extracted 1, 089 records, where anomalies accounted for 5%. Bayesian hyperparameter optimization was employed. We used this strategy with maximizing the AUC metric as the objective and sampled a subset of the data with a fixed size for each experimental run. Hyperparameters $\alpha$, $\gamma$, and $w$ were tuned within the respective ranges of (1, 3), (0.5, 1), and (0.01, 0.05). Subsequently, in the MFL method, $\alpha$, $\gamma$, and $w$ were set to 2.22, 0.61, and 0.049, through Bayesian hyperparameter optimization. To validate the effectiveness of the proposed method on the random forest, we compared it with ensemble methods such as voting, model averaging based on minimizing cross-entropy loss, and commonly used methods such as isolation forest and logistic regression. The dataset was divided into training and test sets in a 70:30 ratio, and the model was trained 60 times. The proposed method was evaluated using the AUC, F1-score, and recall.

Fig 2 illustrates the performance of the model averaging anomaly detection method based on minimizing the MFL criterion on the test set, with average AUC, recall, and F1-score values of 0.8801, 0.7646, and 0.8598, respectively. On the network intrusion dataset, our proposed method achieved the best performance among the tested methods in all metrics, with respective improvements of 0.55%, 5.75%, and 2.90% over the second-best values of AUC, recall, and F1-score, respectively.

To further validate the method's effectiveness in anomaly detection, we selected nine imbalanced datasets from UCI, spanning various domains such as medicine, industrial production, agricultural production, and image classification, and utilized all their features. Simple random sampling was applied to control the ratio of positive to negative samples at 0.05.

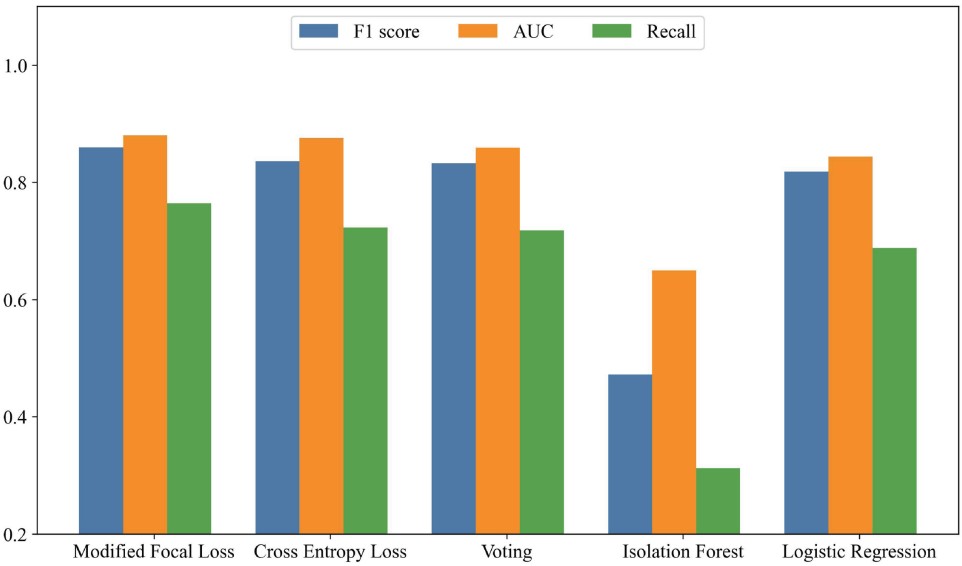

**Fig 2. Results of different models.** Model averaging methods show no significant differences in AUC. Mallows-like method performs well on F1-score and recall, indicating effective detection of outliers.

The datasets were split into training and test sets at a 70:30 ratio. We compared model averaging criteria that minimize different loss functions, along with commonly used outlier detection algorithms such as GMM, Density-Based Spatial Clustering of Applications with Noise (DBSCAN), and LOF [22,23]. The proposed method was evaluated based on AUC, F1-score, and recall. We performed Bayesian hyperparameter tuning. Table 1 presents the results of hyperparameter optimization for the primary outcomes of the model.

Table 2 compares the AUC values of the proposed MFL method and other anomaly detection algorithms. Our model achieves up to an 18.31% improvement over the second-best model across different datasets, and a 9.50% improvement in the mean across nine datasets, which demonstrates its strong data fitting capability. Note that some methods have lost their capacity to effectively distinguish between classes due to the limitations of the methods employed and the severe imbalance in the data. Model averaging methods that use default parameter settings tend to achieve similar fitting performance, highlighting the importance of Bayesian hyperparameter tuning to enhance model performance. Table 3 compares the recall of the MFL method and other anomaly detection algorithms. Recall is the proportion of detected anomalies out of the total number of actual anomalies and hence is crucial

**Table 1. Hyperparameters of different datasets.**

| Dataset | Hyperparameters | | |
|---|---|---|---|
| | $\alpha$ | $\beta$ | w |
| SB | 1.6220 | 0.6626 | 0.0392 |
| Pis | 1.7169 | 0.5579 | 0.0445 |
| MHR | 1.7491 | 0.9754 | 0.0393 |
| PS | 2.6335 | 0.9972 | 0.0436 |
| PCO | 2.9073 | 0.9736 | 0.0434 |
| Ye | 1.1717 | 0.5579 | 0.0445 |
| Ca | 1.7491 | 0.9754 | 0.0393 |
| MF | 1.3920 | 0.5526 | 0.0230 |
| SA | 2.2207 | 0.6107 | 0.0490 |

**Table 2. AUC scores of anomaly detection algorithms.**

| Model \ Dataset | SB | Pis | MHR | PS | PCO | Ye | Ca | MF | Sa | Mean |
|---|---|---|---|---|---|---|---|---|---|---|
| Modified Focal | **0.8702** | **0.7873** | **0.5742** | 0.7944 | **0.8956** | **0.5789** | **0.7500** | **0.9656** | **0.9529** | **0.7966** |
| Focal | 0.7231 | 0.5770 | 0.4951 | 0.7040 | 0.8141 | 0.4964 | 0.6500 | 0.9589 | 0.9134 | 0.7036 |
| Vote | 0.7121 | 0.5694 | 0.4959 | 0.7003 | 0.8118 | 0.4970 | 0.6167 | 0.9560 | 0.9110 | 0.6967 |
| Zero One | 0.7231 | 0.5770 | 0.4951 | 0.7040 | 0.8141 | 0.4964 | 0.6500 | 0.9589 | 0.9134 | 0.7036 |
| Hamming | 0.7231 | 0.5770 | 0.4951 | 0.7040 | 0.8141 | 0.4964 | 0.6500 | 0.9589 | 0.9134 | 0.7036 |
| Hinge Loss | 0.7355 | 0.6201 | 0.5323 | 0.7119 | 0.8032 | 0.5396 | **0.7500** | 0.9498 | 0.9030 | 0.7273 |
| Cross Entropy | 0.7335 | 0.6072 | 0.5388 | 0.7118 | 0.8080 | 0.5383 | **0.7500** | 0.9539 | 0.9064 | 0.7275 |
| Average | 0.7231 | 0.5185 | 0.4950 | 0.7040 | 0.8141 | 0.4964 | 0.6500 | 0.9589 | 0.9134 | 0.7036 |
| IF | 0.5071 | 0.5185 | 0.4604 | 0.7315 | 0.7975 | 0.4400 | 0.6250 | 0.4630 | 0.4232 | 0.5518 |
| Logistic | 0.6914 | 0.5000 | 0.5000 | 0.5000 | 0.5000 | 0.5000 | 0.5000 | 0.5000 | 0.5000 | 0.5213 |
| KNN | 0.5185 | 0.5611 | 0.5000 | 0.5987 | 0.7222 | 0.5000 | 0.5000 | 0.8645 | 0.9277 | 0.6325 |
| GMM | 0.6601 | 0.5483 | 0.5335 | 0.6917 | 0.7852 | 0.3933 | 0.6500 | 0.4349 | 0.4707 | 0.5975 |
| DBSCAN | 0.6976 | 0.000 | 0.5000 | 0.5000 | 0.5000 | 0.5000 | 0.5000 | 0.5000 | 0.5000 | 0.4108 |
| LOF | 0.5285 | 0.7704 | 0.4024 | **0.8195** | 0.8047 | 0.3933 | 0.6500 | 0.4954 | 0.5344 | 0.5998 |
| Improvement (%) | 18.31 | 2.19 | 6.57 | -3.06 | 10.01 | 7.28 | 0.00 | 0.70 | 3.27 | 9.50 |

**Table 3. Recall scores of anomaly detection algorithms.**

| Dataset / Model | SB | Pis | MHR | PS | PCO | Ye | Ca | MF | Sa | Mean |
|---|---|---|---|---|---|---|---|---|---|---|
| Modified Focal | **0.7606** | 0.5993 | 0.1688 | 0.6211 | **0.8025** | 0.1333 | 0.5000 | **0.9321** | **0.9106** | **0.6031** |
| Focal | 0.4470 | 0.1569 | 0.0000 | 0.4203 | 0.6296 | 0.0000 | 0.3000 | 0.9179 | 0.8269 | 0.4110 |
| Vote | 0.4246 | 0.1417 | 0.0000 | 0.4119 | 0.6247 | 0.0000 | 0.2333 | 0.9121 | 0.8221 | 0.3967 |
| Zero One | 0.4470 | 0.1569 | 0.0000 | 0.4203 | 0.6296 | 0.0000 | 0.3000 | 0.9179 | 0.8269 | 0.4110 |
| Hamming | 0.4470 | 0.1569 | 0.0000 | 0.4203 | 0.6296 | 0.0000 | 0.3000 | 0.9179 | 0.8269 | 0.4110 |
| Hinge Loss | 0.4860 | 0.2569 | 0.0813 | 0.4439 | 0.6142 | 0.1083 | **0.5000** | 0.9038 | 0.8078 | 0.4669 |
| Cross Entropy | 0.4470 | 0.1569 | 0.0000 | 0.4203 | 0.6296 | 0.0000 | 0.3000 | 0.9179 | 0.8269 | 0.4110 |
| Average | 0.4470 | 0.1569 | 0.0000 | 0.4203 | 0.6296 | 0.0000 | 0.3000 | 0.9179 | 0.8269 | 0.4110 |
| IF | 0.0227 | 0.1250 | 0.2500 | 0.7783 | 0.7407 | 0.0000 | **0.5000** | 0.0000 | 0.0120 | 0.2699 |
| Logistic | 0.3863 | 0.0000 | **0.5000** | 0.0000 | 0.0000 | 0.0000 | 0.0000 | 0.0000 | 0.0000 | 0.0985 |
| KNN | 0.0455 | 0.1250 | 0.0000 | 0.2000 | 0.4444 | 0.0000 | 0.0000 | 0.7308 | 0.8554 | 0.2668 |
| GMM | 0.4545 | 0.2917 | 0.2500 | 0.5600 | 0.7407 | 0.0000 | 0.5000 | 0.0769 | 0.1446 | 0.3361 |
| DBSCAN | 0.6818 | 0.0000 | 0.0000 | 0.0000 | 0.0000 | 0.0000 | 0.0000 | 0.0000 | 0.0000 | 0.0757 |
| LOF | 0.2045 | **0.7083** | 0.0000 | **0.8000** | 0.7778 | 0.0000 | 0.5000 | 0.1923 | 0.2651 | 0.3609 |
| Improvement (%) | 11.60 | -15.41 | -66.24 | -22.40 | 3.18 | 23.08 | 0.00 | 1.55 | 6.45 | 29.17 |

in anomaly detection. Compared with commonly used model averaging methods and conventional anomaly detection techniques, our method achieves a 29.17% improvement in the mean recall and demonstrates a clear advantage in most scenarios.

Table 4 presents the F1-score value of our proposed method and the comparison models, which serves as a balanced measure of precision and recall. Our approach demonstrates superior performance across most scenarios, highlighting its robust predictive capability with imbalanced datasets.

In the above experiments, we employed simple random sampling to control the positive-to-negative ratio at 1:20. To further demonstrate the applicability of our method and its ability with extremely imbalanced data classification, we conducted experiments on the impact of the positive-to-negative ratio using a network intrusion dataset. Several points were selected, with

**Table 4. F1-scores of anomaly detection algorithms.**

| Dataset / Model | SB | Pis | MHR | PS | PCO | Ye | Ca | MF | Sa | Mean |
|---|---|---|---|---|---|---|---|---|---|---|
| Modified Focal | **0.8522** | 0.7356 | 0.2863 | 0.7500 | **0.8812** | **0.2335** | **0.6562** | **0.9630** | **0.9490** | **0.6886** |
| Focal | 0.6123 | 0.2691 | 0.0000 | 0.5834 | 0.7664 | 0.0000 | 0.4544 | 0.9554 | 0.9014 | 0.4876 |
| Vote | 0.5908 | 0.2464 | 0.0000 | 0.5753 | 0.7628 | 0.0000 | 0.3730 | 0.9522 | 0.8984 | 0.4847 |
| Zero One | 0.6123 | 0.2691 | 0.0000 | 0.5834 | 0.7664 | 0.0000 | 0.4544 | 0.9554 | 0.9041 | 0.4876 |
| Hamming | 0.6123 | 0.2691 | 0.0000 | 0.5834 | 0.7664 | 0.0000 | 0.4544 | 0.9554 | 0.9041 | 0.4876 |
| Hinge Loss | 0.6453 | 0.4034 | 0.1497 | 0.6045 | 0.7528 | 0.1944 | **0.6562** | 0.9456 | 0.8889 | 0.5587 |
| Cross Entropy | 0.6117 | 0.2691 | 0.0000 | 0.5828 | 0.7652 | 0.0000 | 0.4565 | 0.9537 | 0.9007 | 0.4875 |
| Average | 0.6123 | 0.2691 | 0.0000 | 0.5834 | 0.7664 | 0.0000 | 0.4544 | 0.9554 | 0.9014 | 0.4876 |
| IF | 0.0443 | 0.2184 | 0.3613 | 0.7317 | 0.7908 | 0.0000 | 0.5926 | 0.0000 | 0.0236 | 0.3187 |
| Logistic | 0.5519 | 0.0000 | **0.5000** | 0.0000 | 0.0000 | 0.0000 | 0.0000 | 0.0000 | 0.0000 | 0.1225 |
| KNN | 0.0868 | 0.2207 | 0.0000 | 0.3304 | 0.6097 | 0.0000 | 0.0000 | 0.8394 | 0.9187 | 0.3144 |
| GMM | 0.5940 | 0.6420 | 0.3920 | 0.6620 | 0.7800 | 0.0000 | 0.6250 | 0.1440 | 0.2530 | 0.4547 |
| DBSCAN | 0.6960 | 0.6520 | 0.0000 | 0.0000 | 0.0000 | 0.0000 | 0.0000 | 0.0000 | 0.0000 | 0.1508 |
| LOF | 0.3182 | **0.7960** | 0.0000 | **0.8170** | 0.8020 | 0.0000 | 0.6250 | 0.3240 | 0.4000 | 0.4876 |
| Improvement (%) | 22.44 | -7.59 | -42.74 | -8.20 | 21.14 | 20.10 | 0.00 | 1.55 | 6.45 | 23.25 |

positive-to-negative ratios ranging from 1:10 to 1:100. The evaluation was based on AUC and recall.

We first analyzed the impact of the positive-to-negative ratio on AUC and recall. Fig 3 shows the results of experiments conducted with ratios of 0.01, 0.02, 0.03, 0.05, 0.07, and 0.10. Model averaging methods generally outperform individual anomaly detection models. When the ratio is between 0.05 and 0.10, our proposed method shows a significant advantage in AUC and recall. For ratios of 0.02 and 0.03, all model averaging methods perform approximately the same. We speculate that the performance advantage is due to the tree structure of the base models. As the ratio further decreases, logistic regression fails to effectively distinguish between classes. Overall, our proposed method performs well under highly imbalanced conditions.

### 3.2 Synthetic data experiment

In this experiment, we employed a data sampling method to control the proportion of sample points. However, such an approach may lead to distribution shifts in the data, thereby reducing the reliability of results. To address this issue, we utilized Adaptive Synthetic Sampling [24] to regenerate data points from the sampled data. Using the classic Spambase dataset as an example, the primary performance metrics of various models based on both original and augmented data are summarized in Table 5.

Through the application of the ADASYN method, it is observed that the performance of most models has been enhanced. This indicates that ADASYN effectively synthesizes valuable data points, thus improving model performance. It is worth noting that, despite a reduction in the margin by which our model leads, it still achieves the best performance among the tested methods, which suggests that the improvements in our approach stem from innovations in the model architecture rather than shifts in data distribution and also that our method can robustly handle such distributional shifts.

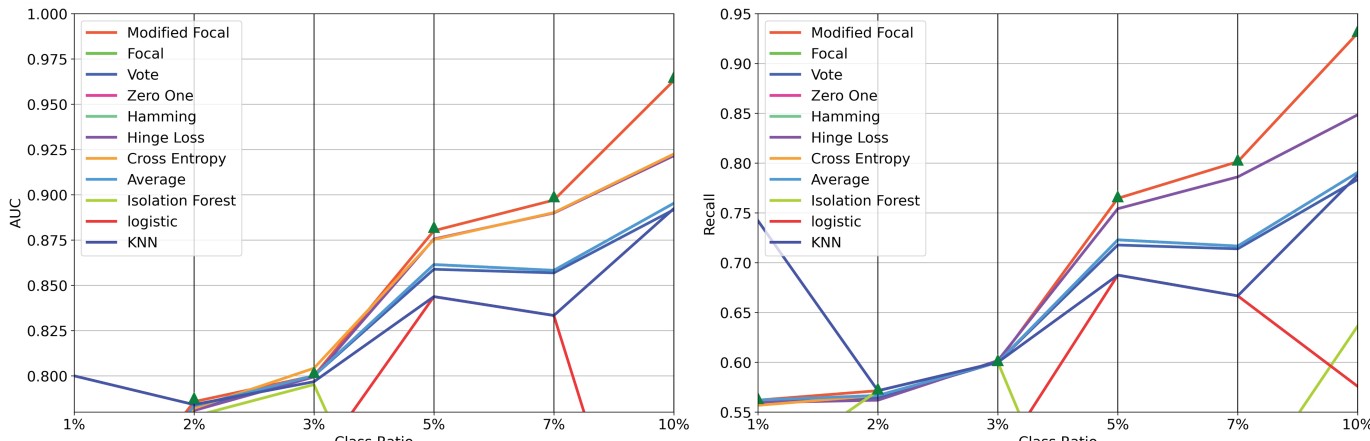

**Fig 3. Impact of class imbalance.** A: AUC; B: recall. Mallows-like loss function demonstrates superior predictive performance across multiple scenarios, indicating significant advantages of model averaging methods based on this loss function.

**Table 5. Experimental results after data augmentation using ADASYN.**

| Dataset / Model | Original | | | Augmented | | |
|---|---|---|---|---|---|---|
| | recall | AUC | F1-score | recall | AUC | F1-score |
| Modified Focal | **0.7606** | **0.8702** | **0.8522** | **0.8811** | **0.9056** | **0.9034** |
| Focal | 0.4470 | 0.7231 | 0.6123 | 0.6947 | 0.8395 | 0.8256 |
| Vote | 0.4246 | 0.7121 | 0.5908 | 0.6845 | 0.8354 | 0.8264 |
| Zero One | 0.4470 | 0.7231 | 0.6123 | 0.6947 | 0.8395 | 0.8256 |
| Hamming | 0.4470 | 0.7231 | 0.6123 | 0.6947 | 0.8395 | 0.8256 |
| Hinge Loss | 0.4860 | 0.7355 | 0.6453 | 0.6799 | 0.8259 | 0.8134 |
| Cross Entropy | 0.4470 | 0.7355 | 0.6117 | 0.6947 | 0.8356 | 0.8234 |
| Average | 0.4470 | 0.7231 | 0.6123 | 0.6947 | 0.8395 | 0.8256 |
| IF | 0.0227 | 0.5071 | 0.0443 | 0.0000 | 0.4934 | 0.0000 |
| Logistic | 0.3863 | 0.6914 | 0.5519 | 0.8636 | 0.8905 | 0.8714 |
| KNN | 0.0455 | 0.5185 | 0.0868 | 0.5000 | 0.6996 | 0.6375 |
| GMM | 0.6601 | 0.4545 | 0.5940 | 0.2955 | 0.5764 | 0.4324 |
| DBSCAN | 0.6976 | 0.6818 | 0.6960 | 0.6818 | 0.6976 | 0.6960 |
| LOF | 0.5285 | 0.2045 | 0.3182 | 0.3409 | 0.5739 | 0.4824 |
| Improvement (%) | 9.03 | 25.87 | 22.44 | 2.03 | 1.70 | 3.67 |

## 4 Conclusion and future work

We proposed a Mallows-like model averaging criterion for anomaly detection based on the focal loss function. This criterion was implemented in a random forest algorithm to address the occurrence of extremely imbalanced data. We compared our method with other ensemble models, as well as commonly used anomaly detection methods, on public benchmark datasets. The results indicated the superior performance of our method in terms of anomaly recall and classification accuracy.

In the future, a Mallows-like focal loss criterion in heteroscedasticity could be investigated, possibly replacing the variance term in the Mallows criterion with the residuals for each sample. The performance of the proposed approach is constrained by the selection of hyperparameters in focal loss. A possible future research direction involves the design of efficient hyperparameter tuning methods tailored for anomaly detection algorithms and theoretical support for classification asymptotic optimality.

## 5 Supporting information

**S1 Table**. **Hyperparameters of different datasets.**
(PDF)

**S2 Table**. **AUC scores of anomaly detection algorithms.**
(PDF)

**S3 Table**. **Recall scores of anomaly detection algorithms.**
(PDF)

**S4 Table**. **F1-scores of anomaly detection algorithms.**
(PDF)

**S5 Table**. **Experimental results after data augmentation using ADASYN.**
(PDF)

**S1 Fig. Schematic diagram of proposed model averaging method.**
(TIF)

**S2 Fig. Results of different models.**
(TIF)

**S3 Fig. Impact of class imbalance.**
(TIF)

## Author contributions

**Conceptualization:** Gaoxiang Zhao.

**Data curation:** Gaoxiang Zhao.

**Methodology:** Gaoxiang Zhao, Lu Wang, Xiaoqiang Wang.

**Project administration:** Xiaoqiang Wang.

**Resources:** Xiaoqiang Wang.

**Supervision:** Xiaoqiang Wang.

**Validation:** Gaoxiang Zhao.

**Visualization:** Gaoxiang Zhao.

**Writing – original draft:** Gaoxiang Zhao.

**Writing – review & editing:** Gaoxiang Zhao, Xiaoqiang Wang.

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
