## [Decision Letter · Decision Letter 0]

PONE-D-24-52217A Mallows-like Criterion for Anomaly Detection with Random Forest ImplementationPLOS ONE

Dear Dr. Wang,

Thank you for submitting your manuscript to PLOS ONE. After careful consideration, we feel that it has merit but does not fully meet PLOS ONE’s publication criteria as it currently stands. Therefore, we invite you to submit a revised version of the manuscript that addresses the points raised during the review process.

Please revise the paper by addressing the reviewer's comments.

We look forward to receiving your revised manuscript.

Kind regards,

Jie Zhang

Academic Editor

PLOS ONE

Journal Requirements:

4. We note that there is identifying data in the Supporting Information file <dataset.zip>. Due to the inclusion of these potentially identifying data, we have removed this file from your file inventory. Prior to sharing human research participant data, authors should consult with an ethics committee to ensure data are shared in accordance with participant consent and all applicable local laws.

-Location data

Reviewers' comments:

Reviewer's Responses to Questions

**Comments to the Author**

1. Is the manuscript technically sound, and do the data support the conclusions?

Reviewer #1: Yes

2. Has the statistical analysis been performed appropriately and rigorously? 

Reviewer #1: Yes

3. Have the authors made all data underlying the findings in their manuscript fully available?

Reviewer #1: Yes

4. Is the manuscript presented in an intelligible fashion and written in standard English?

Reviewer #1: Yes

5. Review Comments to the Author

Reviewer #1: Reviewer Comments on a Mallows-like Criterion for Anomaly Detection with Random Forest Implementation

1. The researchers mainly rely on Focal loss to address class imbalance, but I suggest testing other approaches (e.g., cost-sensitive learning or ensemble techniques such as ADASYN) to prove the quality of the method used, as using Focal loss may lead to certain drawbacks if the synthetic data points do not represent the actual minority class distribution.

2. The paper does not adequately mention how the features were selected, as the performance of machine learning models depends heavily on the quality of the input features.

3. Why did the researchers use mallow-like Criterio in particular? Are there specific characteristics of the dataset that make this more suitable than others?

4. Authors are advised to be precise in the abstract, and structure your abstract as follows- 1) Background 2) Aim/Objective 3) Methodology 4) Results 5) Conclusion. Write 2-4 lines for each and merge everything in one paragraph (200-300 Words) without any subheading.

5. The results were based on AUC, ARI, accuracy and recall. However, there are other important metrics for imbalanced datasets, such as F1 score and precision, which are very important in situation where false negatives are costly.

6. Authors should explain why the choice of Hierarchical sampling over other methods.

7. Authors should state the hyperparameters and values of Bayesian model used in the research for hyperparameter tuning.

8.Authors should create a new section in the experimental analysis section and compared the findings of the proposed model with existing works in the literature.

9. Incorporating relevant and recent academic sources could strengthen your paper’s validity and give readers more context and background. Please avoid citing sources that were published before 2019. Cite current research that are really pertinent to your topic. The study also lacks sufficient citations. Authors can use and depend on these essential works while addressing the topic of their paper and current issues. Some latest papers listed below which studied similar problems can be added: 1). XIDINTFL‑VAE: XGBoost‑based intrusion detection of imbalance network traffic via class‑wise focal loss variational autoencoder. 2). Feature selection in intrusion detection systems: a new hybrid fusion of Bat algorithm and Residue Number System. 3). Towards an efficient model for network intrusion detection system (IDS): systematic literature review. 4). GA-mADAM-IIoT: A new lightweight threats detection in the industrial IoT via genetic algorithm with attention mechanism and LSTM on multivariate time series sensor data.

10. In the end of related work section, highlight in 10-15 lines what overall technical gaps are observed in existing techniques that led to the design of the proposed methodology.

11. Although the English is generally quite good, there are quite a few minor grammatical errors, and a careful read-through is needed to eliminate these errors. The spelling mistake should be corrected by reading through the manuscript.

12. Please Change the “conclusion” section to “Conclusion and Future Work” and write future work.

6. PLOS authors have the option to publish the peer review history of their article (what does this mean?). If published, this will include your full peer review and any attached files.

Reviewer #1: No

---

## [Author Response · Author response to Decision Letter 1]

26 Feb 2025

Thank you very much for your valuable suggestions. We have made detailed revisions to the article content according to the reviewers' comments, and the revisions have been included in the file "Response to Reviewers.pdf".

---

## [Decision Letter · Decision Letter 1]

A Mallows-like Criterion for Anomaly Detection with Random Forest Implementation

PONE-D-24-52217R1

Dear Dr. Wang,

We’re pleased to inform you that your manuscript has been judged scientifically suitable for publication and will be formally accepted for publication once it meets all outstanding technical requirements.

Kind regards,

Jie Zhang

Academic Editor

PLOS ONE

Additional Editor Comments (optional):

Reviewers' comments:

Reviewer's Responses to Questions

**Comments to the Author**

1. If the authors have adequately addressed your comments raised in a previous round of review and you feel that this manuscript is now acceptable for publication, you may indicate that here to bypass the “Comments to the Author” section, enter your conflict of interest statement in the “Confidential to Editor” section, and submit your "Accept" recommendation.

Reviewer #1: All comments have been addressed

2. Is the manuscript technically sound, and do the data support the conclusions?

Reviewer #1: Yes

3. Has the statistical analysis been performed appropriately and rigorously? 

Reviewer #1: Yes

4. Have the authors made all data underlying the findings in their manuscript fully available?

Reviewer #1: Yes

5. Is the manuscript presented in an intelligible fashion and written in standard English?

Reviewer #1: Yes

6. Review Comments to the Author

Reviewer #1: (No Response)

7. PLOS authors have the option to publish the peer review history of their article (what does this mean?). If published, this will include your full peer review and any attached files.

Reviewer #1: **Yes: **Yakub Kayode Saheed

---

## [Editor Report · Acceptance letter]

PONE-D-24-52217R1

PLOS ONE

Dear Dr. Wang,

I'm pleased to inform you that your manuscript has been deemed suitable for publication in PLOS ONE. Congratulations! Your manuscript is now being handed over to our production team.

Kind regards,

on behalf of

Dr. Jie Zhang

Academic Editor

PLOS ONE